# Application of Three Compounds Extracted from *Cynodon dactylon* against *Streptococcus mutans* Biofilm Formation to Prevent Oral Diseases

**DOI:** 10.3390/biom13091292

**Published:** 2023-08-24

**Authors:** Tasnia Habib, Aminur Rahman, Anroop B. Nair, S. M. Shahinul Islam

**Affiliations:** 1Udayan Dental College, Rajshahi 6000, Bangladesh; tasniahabibsinthia@gmail.com; 2Department of Biomedical Sciences, College of Clinical Pharmacy, King Faisal University, P.O. Box 400, Al-Ahsa 31982, Saudi Arabia; 3Department of Pharmaceutical Sciences, College of Clinical Pharmacy, King Faisal University, Al-Ahsa 31982, Saudi Arabia; 4Plant Biotechnology and Genetic Engineering Laboratory, Institute of Biological Sciences, University of Rajshahi, Rajshahi 6205, Bangladesh

**Keywords:** *Streptococcus mutans*, biofilm, *Cynodon dactylon*, oral diseases, bacterial growth inhibition

## Abstract

*Streptococcus mutans* bacteria form a biofilm called plaque that causes oral diseases, including tooth decay. Therefore, inhibition of biofilm formation is essential to maintaining good oral health. The health and nutritional benefits of *Cynodon dactylon* are well documented, but very little is known about its use to treat against oral diseases. The aim of this study was to detect the adhesion strength of the *S. mutans* bacterial biofilm in 100 cases in the Rajshahi region and evaluate the inhibitory activity of different compound extracts of *C. dactylon* on the *S. mutans* bacterial biofilm by determining the composition of isolated compounds using phytochemical analysis. Nuclear magnetic resonance (NMR) spectroscopy confirmed that three specific compounds from *C. dactylon* were discovered in this study: 3,7,11,15 tetramethyl hexadec-2-4dien 1-o1, compound 3,7,11,15 tetramethylhexadec-2-en-1-o1 from phytol derivatives, and stigmasterol. Results indicated that the compound of 3,7,11,15-tetramethyl-hexadec-2-en-1-ol exhibited higher antibiofilm activities on *S. mutans* than those of the other compound extracts. A lower level of minimum inhibitory concentration was exposed by 3, 7, 11,15 tetramethyl hexadeca-2-en-1-o1 (T2) on *S. mutans* at 12.5 mL. In this case, the compound of 3,7,11,15 tetramethyl hexadec 2en-1-o1 was used, and patients showed a mean value and standard error reduced from 3.42 ± 0.21 to 0.33 ± 0.06 nm. The maximum inhibition was (80.10%) in the case of patient no. 17, with a value of *p* < 0.05 found for *S. mutans* to which 12.5 μL/mL ethyl acetate extract was applied. From these findings, it may be concluded that *C. dactylon* extracts can be incorporated into various oral preparations to prevent tooth decay.

## 1. Introduction

Microbial biofilms are intricate bacterial colonies in the environment and the human body. Oral infectious diseases are the biggest problems of dental health care and are intimately related to microbial colonization and the development of pathogenic biofilms such as pulp infection, tooth decay (dental caries), periodontitis, induced wound infection, peri-implantitis, and tooth staining [1,2,3,4]. More than 3.5 billion people are affected by oral disorders, which significantly lower the quality of life, particularly in many low- and middle-income nations [1,5,6,7,8,9]. Dental plaque has been recognized as a biofilm in recent years [10]. Oral biofilm is the three-dimensional arrangement of various microorganisms residing in the mouth cavity. All tissues and surfaces in the mouth cavity can be adhered to by bacterial biofilms. Human tooth decay and the inflammatory degeneration of the alveolar bone are caused by oral biofilms [11]. If the biofilm remains untreated, it can undergo maturation, leading to the development of dental caries [10]. Dental caries is a long-lasting disease that leads to the demineralization of enamel, dentin, and cementum. Dental caries can be brought on by four things: the host, substrate, bacteria, and time. The interactions of these four factors impact the development of dental caries. Bacterial biofilms are observed on oral mucous membranes, tooth surfaces, and dental prosthetic items in regions connected to oral health care. The causative agent in dental caries and periodontal illnesses is a biofilm, which manifests as supragingival and subgingival plaque [10]. Oral biofilms are multispecies microbial colonies that show strong antibiotic resistance [12]. Dental caries development is associated with biofilm formation, affecting a large population worldwide [13]. Dental plaque causes biofilm-related oral infections, which are common and expensive and can result in caries or other oral illnesses.

Bacterial biofilms mostly bring on pathogenic processes in the oral environment. More than 700 bacteria can be found in the mouth cavity [14]. Among them, the most common bacteria in the mouth are streptococci; notably, *Streptococcus mutans* bacterial species are the preliminary etiologic agent for biofilm formation [15]. *S. mutans* bacteria also play an essential role in informing this multidimensional process, and these microorganisms can form biofilm structures and produce organic acids [16]. *S. mutans* bacteria utilize the acidic metabolites produced by carbohydrates, leading to acidic destruction of the tooth enamel, which causes dental caries [17]. *S. mutans* bacteria are acidophilic in nature and can survive at low pH levels and synthesize glucan as part of plaque formation [18]. *S. mutans*, among all bacterial species, is thought to have a significant part in complex structure generation, creating multidimensional regions on the oral mucosa and tooth surfaces [19]. *S. mutans* bacteria produce intracellular iodine-staining polysaccharides (IPS) from sucrose, which frequently resembles glycogen. IPS can be metabolized, producing acids continuously [20]. Due to their intracellular iodine-staining polysaccharide storage, these cariogenic bacteria may continue fermenting sugar without exogenous food sources [21]. The essential requirement for developing dental caries related to *S. mutans* is protein–bacterium interaction with the involvement of sucrose-dependent and independent mechanisms [13].

So, after analyzing all the data, it is confirmed that *S. mutans* bacteria is the most pathogenic microorganism found in biofilms other than any other oral streptococci. Without the colonization and the progression of *S. mutans* bacterial biofilm, many oral diseases such as dental caries, gingivitis, or periodontitis cannot be developed. So, preventing the formation of *S. mutans* biofilm is mandatory for restricting the development of many oral diseases. In addition, developing inexpensive and efficient multifunctional preventive and treatment methods is also essential.

Chemical substances such as antimicrobial agents might be used to achieve mechanical plaque control. In addition to being expensive and ineffective for treating with chemical antimicrobial agents, these may have adverse effects. The development of drug-resistant pathogens, primarily involved in nosocomial infections, has raised concern among medicinal practitioners [22]. It also created a problem in controlling the growth of infectious diseases caused by pathogens. Hence, looking for novel infection-fighting techniques to control and prevent microbial infections is necessary. Given the alarming incidence of antibiotic resistance in pathogens, which raises concern among medical practitioners, there is a constant need for new and effective therapeutic agents. Therefore, it is essential to develop an alternative antimicrobial medicine to treat infectious diseases using medicinal plants [23]. The creation of novel therapeutic compounds must pay particular attention to microbial resistance to conventional antibiotics. Medicinal plants play an essential role in the pharmaceuticals industry in developing alternative drugs to overcome the drawback influenced by synthetic drugs [24]. For this study, the medicinal plant *Cynodon dactylon* was considered, which belongs to the family of Poaceae and is found almost everywhere on Earth [25]. *C. dactylon* is also known as Bermuda grass or Doob grass. It is divided into three parts: the root, stem, and leaves. It is fast-growing and readily available everywhere. Furthermore, it is commonly referred to in Bengali as Dub, Durva, Dubla, Durba, and Neel Doorva. It has many medicinal activities like antidiabetic, diuretic, antioxidant, and allergic effects [26] and is used as a rejuvenator for wound healing [27]. Plants like *C. dactylon* contain a large number of chemicals like organic compounds, e.g., alkaloids, flavonoids, glycosides, ß-sitosterol, and carotene, stigmasterol, phytol, fryer phenols, etc. The antibacterial activities of *C. dactylon* are primarily attributed to a particular extract that contains quinines, tannins, and phenols. Phenolic aldehydes are used as a secondary metabolite that acts as a defensive mechanism against respective pathogens [26,27] and acts as a bioactive, chemotherapeutic, bactericidal, and bacteriostatic agent [28]. Thus, one of the key goals of dental practitioners, essential to maintaining excellent oral health, is control of dental plaque biofilm. This study also aimed to determine the antimicrobial and antibiofilm effectiveness of different compounds of *C. dactylon* against the biofilm of *S. mutans* because dental plaque cannot be developed without the growth of these bacteria.

## 2. Materials and Methods

### 2.1. Culture Media

Mitis Salivarius Base (MSB) agar and Luria Bertani (LB) media were used to selectively isolate and identify *Streptococcus mutans* selectively. First, 90.07 g MSB agar base powder was suspended in 1000 mL of distilled water, heated to boil it properly, and sterilized by autoclaving at 121 °C for 15 min. Next, the MSB media was cooled to 50–55 °C, 1 mL of sterile potassium tellurite solution was added, and the mixture was poured into sterile Petri plates.

### 2.2. Collection of Dental Plaque Samples from Patients

Dental plaque samples were collected from 100 individuals having various oral complications patients attending the outdoor Department of the Dental Unit at Rajshahi Medical College and Hospital, Bangladesh. As part of these 100 individuals, 32 children (approximately 1 to 10 years old) 41 teenagers (11 to 18 yearsrs old), and 27 adults were selected for this study as the WHO (World Health Organization) reported that 60 to 90% of children and young adults affected by dental caries have *S. mutans* bacterial biofilm to a greater extent than adults [29]. Therefore, among all the 100 cases, 2 (case numbers 17 and 55) were selected for further analysis. The plaques were spread on each Petri plate containing MSB selective media and incubated at 37 °C for 72 h.

### 2.3. Formation of S. mutans Bacterial Biofilm by Microtiter Plate and Microtiter Spectrophotometric Biofilm Production Methods

The biofilm assay was performed following the O’Toole protocol with slight changes [30]. Briefly, four to five mL liquid LB medium was taken to each test tube, and a single colony of *S. mutans* was added and incubated at 37 °C for 24 h at 120 rpm. The overnight culture was diluted at 1:100 into fresh LB medium for biofilm assays. Then, the cultures were distributed in flat-bottom 96-well polystyrene microtiter plates in triplicate at 200 μL per well (Figure 1). Two other strains named *Pseudomonas aeruginosa* and *Escherichia coli* were selected for control samples in this study for biofilm formation. *P. aeruginosa* is a Gram-negative aerobic non-spore-forming rod and *E. coli* is a facultative Gram-negative rod-shaped bacterium. These two bacteria were collected from the stock bacterial culture of the molecular pathology laboratory of the Institute of Biological Sciences, Rajshahi University. The microtiter plate was incubated at 37 °C for 24 h. After incubation, planktonic cells were aspirated and washed three times with sterile saline (0.9%). The biofilms were fixed with 99% methanol for 15 min and were stained with 200 µL of 0.1% crystal violet for 15 min. The excess crystal violet was drained and biofilms were washed thrice with tap water and air dried. Finally, the adherent cell-bound crystal violet was dissolved by adding 200 μL of 33% glacial acetic acid to each well, and biofilms were quantified using a microplate reader at OD_595_.

### 2.4. Collection and Preparation of Cynodon dactylon

*Cynodon dactylon* was collected from the Botanical Garden of Rajshahi University campus, Rajshahi, Bangladesh, and my supervisor confirmed the conformity of collected plants. *C. dactylon* was thoroughly washed with tap water, rinsed with distilled water, and air dried in a shady place. Finally, it was pulverized to fine powder using a grinding machine. Ethyl acetate was added with the pulverized *C. dactylon* and filtrated using Whatman 0.45 µm filter paper. The filtrated crude extracts were dried and stored for further use.

### 2.5. Isolation and Purification of Compounds from C. dactylon

This study was conducted at the Insect Biotechnology and Molecular Pathology Laboratory, Institute of Biological Sciences, Rajshahi University, and Rajshahi Dental College. The compounds were isolated and purified using column chromatographic techniques, following the protocol described in the flow chart (Figure 2). The dry columns were prepared with silica gel and solvents such as n-hexane, acetone, ethyl acetate, etc., were used. The extracted compounds were identified using gas chromatography–mass spectrometry (GC–MS) analysis, and structure determination was conducted by using nuclear magnetic resonance (NMR) analysis. Here, we provide the spectral data of 1H NMR and C-13 NMR spectroscopy with the chemical structure of the compound of 3,7,11,15 tetramethyl hexadec-2,4 dien 1-o1, 3,7,11,15 tetramethyl hexadec 2en-1-o1 from phytol derivatives, and stigmasterol according to Ahmed et al. [31] as well as Subavathy and Thilaga [32].

### 2.6. Prevention of S. mutans Biofilm Using Different Compounds Extracted from C. dactylon

Biofilms were produced from bacteria grown on MSB medium using a microplate-based system in an in vitro model, which detected the adhesion strength of the biofilm developed by *S. mutans* in the 100 studied samples. Case 17, a 9-year-old, had the greatest adhesive forces with the underlying substratum of LB media (4.12 ± 0.31), while case 55, a 35-year-old, had the greatest adhesive forces of *S. mutans* bacterial biofilm with the substratum of LB media with the mean value 4.32 ± 0.19. Therefore, these two cases were selected to observe the inhibitory effects of the three compounds derived from ethyl acetate extract. Minimum inhibitory concentration (MIC) of the extracts was determined using a broth dilution technique as described by Adikwu et al. [28]. Twofold serial dilutions of the extracts were prepared by adding 5 mL of 100 mg/mL of the extract into a test tube containing 5 mL of nutrient broth and mixed vigorously, thus producing a solution containing 50 mg/mL of the extract. The process continued serially up to the fifth test tube and the last 5 mL was discarded, leaving equal volume in the tubes, hence producing the following concentrations: 50, 25 and 12.5 mg/mL. McFarland standards of test organisms were introduced into the disc and incubated at 37 °C for 24 h.

### 2.7. Statistical Analysis

All the information is stated as mean ± SE (standard error) of the mean. Statistical analysis was performed using SPSS software (version 16). The significance of differences and comparisons among the mean values were determined by Duncan’s multiple range tests (DMRT) as well as the *p*-value reached from the DMRT test at a 1% level (*p* < 0.01). The isolated compound was determined using nuclear magnetic resonance (NMR) spectroscopy and the microtiter spectrophotometric biofilm method measured the adhesion strength (nanometer).

## 3. Results

### 3.1. Microbiological Investigation

Dental plaques and saliva samples were collected from patients from the dental unit of Rajshahi Medical College and Hospital, spread on an MSB (mitis salivarius bacitracin) medium, and inoculated with long-chain purple-colored *Streptococcus mutans* bacterial colonies. The isolates showed Gram-positive *Cocci* in chains with hard, raised, convex pale blue colonies with a frosty glass appearance (Figure 3).

### 3.2. Extraction of Compounds from C. dactylon

The crude extract of *C. dactylon* and purified products were run through thin-layer chromatography. After finishing this procedure, several compound bands are visualized on the TLC plate after being eluted with a mixture of two solvents sprayed with vanillin–sulfuric acid reagents. These compounds were run through on the TLC plate to obtain specific bands of the molecules. The compounds’ names were confirmed by the spectral data of 1H NMR and C-13 NMR spectroscopy. The chemical structures of the three compounds were compared by gas chromatography–mass spectrometry (GC–MS) analysis. The results are displayed in Figure 4 and Figure 5, respectively.

Figure 5 shows that each lane contains several brands expressing a mixture of several compounds. These compounds were further run through TLC to obtain the specific brands of the molecules. Figure 5 displays the three compounds obtained from the ethyl acetate extract of *C. dactylon*—3, 7,11,15 tetramethylhexdec-2,4 dien-1-o1 (Figure 5A) as a dark blue color, 3,7,11,15 tetramethylhexadec-2en-1-o1 from phytol derivatives (Figure 5B) as a light blue color, and stigmasterol (Figure 5C) as a green color—after being subjected to the solvent system in n-hexane:acetone (7:2) sprayed with a vanillin–sulfuric acid reagent.

The isolated compounds were subjected to several analyses, including different spectroscopic analyses such as ^1^H nuclear magnetic resonance (NMR) and C-13 NMR spectroscopy, accomplished by utilizing Bruker 600 and 400 MHz instruments and reported in CDCl3, recorded on a Bruken advance 11,400 NMR spectrometer at the Department of Microbiology, Jahangirnagar University, Bangladesh. CDCl3 is a trichloromethane (chloroform) molecule in which the hydrogen has been replaced by its isotope deuterium and is commonly used as the solvent in proton NMR. NMR signals are analyzed with respect to two characteristics such as intensity and frequency (megahertz (MHz) are usually used for the measurement of absolute frequencies); all of the data are presented in Figure 6, Figure 7 and Figure 8. After completion of structure determination by NMR spectroscopy, it was confirmed that the active compounds were 3,7,11,15 tetramethyl hexadec-2,4 dien 1-o1 (T_1_), 3,7,11,15 tetramethyl hexadec 2en-1-o1 (T_2_) from phytol derivatives, and stigmasterol (T_3_) following the standard protocol of Ahmed et al. [31] as well as that of Subavathy and Thilaga [32] by gas chromatography–mass spectrometry (GC–MS) analysis.

### 3.3. Spectral Analysis of the Compounds from C. dactylon

The three isolated compounds from the ethyl acetate extract of C. dactylon are shown in Figure 6. Analysis of the ChemDraw spectrum of the phytol derivative compound (3, 7, 11, 15 ttramethylhexadeca-2-4 dien 1-01) showed six olefinic protons at δ 5.15, 5.03, and 5.96 and 5.03, 5.70, and 4.97, corboxic acid at 11.0 ppm as the primary amine proton, and δ 2.0 generally disposed the methyl proton at δ 1.33 as a singlet. Figure 7 shows the analysis of the phytol derivative compound 3, 7, 11, 15 tetramethyhexadeca 2-en-1-o1; six olefinic protons were revealed, similar to compound 3, 7, 11, 15 tetramethyhexadeca-2-4 dien 1-o1, and two methyl protons were observed at δ 3.23 and 3.24.

The 1H and C-13 NMR spectra of these two compounds are presumed to indicate all peaks present in these two compounds of phytol derivatives with some impurities. In the case of Figure 8, we revealed that the C-13 NMR spectrum of stigmasterol showed recognizable signals at 19.064, 40.543, 121.321, 129.341, 138.404, and 140.943 ppm. Finally, the nomenclature of these above three compounds was confirmed by comparing it with some data obtained from a previous study conducted by Ahmed et al. [31] as well as Subavathy and Thilaga [32] using gas chromatography–mass spectrometry (GC–MS) analysis. The main compound is 3, 7, 11, 15- tetramethyl-headeca-2-en-1-o1 (Table 1).

### 3.4. Patient Status with Dental Plaque and Oral Hygienic Grounds for Biofilm Formation

For biofilm formation, 100 cases were enumerated and their dental plaque and oral hygienic conditions were observed. Among these 100 cases, there were females 44 and 56 males (Appendix A). All the patients’ ages were divided into six groups. So, among these 100 patients, 28 were 5 to 10 years old, 23 were 11 to 16 years old, 17 were 17 to 22 years old, 14 were 23 to 28 years old, 10 were 29 to 34 years old, and 8 were 35 to 40 years old. In terms of dental plaque, mild, moderate, and severe scores were obtained. This study included patients with different oral hygiene agreements such as good, average, and poor. Of the patients, 10 cases (case numbers 4, 17, 24, 33, 55, 63, 68, 80, and 93) had severe plaque; approximately 38 cases had moderate plaque, and the rest of cases (52) had mild plaque.

The majority of patients (61%) had good oral hygiene, while 31% had average and 8% had poor oral hygiene. The poor-oral-hygiene patients had allocated case numbers 4, 7, 17, 33, 55, 63, 68, and 80 (Appendix A). Case number 17 was a 9 year-old-male patient; on the other hand, case number 55 was a 35-year-old female patient (Appendix A).

### 3.5. Biofilm Formation of Dental Plaque-Causing Bacteria

The adhesion strength of *S. mutans* bacterial biofilm was quantified for all 100 cases and the results are displayed in Figure 9. The results showed that the biofilm formation of all the patients was divided into three groups: weak, moderate, and strong. The biofilm formation ranges from 0.5 to 2.0 nm for the weak group, 2.0 to 3.5 nm for the moderate group, and 3.5 to 5.0 nm for the strong group (Table 2). Among the 100 patients, 54 (54%) were in the weak group, 34 (34%) were in the moderate group, and 12 (12%) were in the strong group for biofilm formation. The mean value of biofilm formation was 3.97 ± 0.46 for the strong group, followed by 2.79 ± 0.31 for the moderate group and 1.41 ± 0.53 for the weak group, showing significant differences (Table 2). The results show that patient numbers 17 and 55 were confirmed to have presented the highest level of biofilm formation and strong biofilm formation, respectively. Patient number 17 exhibited the highest level (4.48 ± 0.42 nm) of biofilm formation. Patient number 55 showed 4.32 ± 0.65 nm (Figure 9). Thus, patients 17 and 55 were selected for biofilm formation.

### 3.6. Effect of the Plant Extracts on the Growth of Biofilm Formation

After extracting the compounds from *C. dactylon*, the three compounds were applied for minimum inhibitory concentration (MIC), as denoted in Table 3. The results show that dilutions of the three compounds derived from ethyl acetate extract (3, 7, 11,15 tetramethyl hexadeca-2,4 dien1-o1 (T_1_), 3, 7, 11,15 tetramethyl hexadeca-2-en-1-o1 (T_2_), and stigmasterol (T_3_)) can inhibit the segregates on the biofilm produced by three bacteria viz. *P. aeruginosa*, *E. coli,* and *S. mutans*. A lower minimum inhibitory concentration was exposed by 3, 7, 11,15 tetramethyl hexadeca-2-en-1-o1 (T_2_) on *S. mutans* at 12.5 µL/mL.

The results were compared with the control (without adding any compounds) and are demonstrated in Figure 10. The results showed that all three compounds possess antimicrobial activities against *S. mutans, E. coli,* and *P. aeruginosa* bacterial biofilm or dental plaque. The utmost motion showed evidence of *Streptococcus mutans* for 12.5 μL of the extract. Compound 3, 7, 11, 15-tetramethyl-hexadeca-2-en-1-ol (T_2_) was confirmed to possess peak antimicrobial properties in favor of the escalation of reticence on top of the biofilm of *Streptococcus mutans* along with the other compounds of *Cynodon dactylon*. The compound 3, 7, 11, 15 tetramethyl hexadec 2en-1-o1 (T_2_) showed a maximum inhibition of 3.42 ± 0.21 nm for case No. 17 (Figure 10A) and 3.07 ± 0.14 nm for case No. 55 (Figure 10B) in *Streptococcus mutans*. On the other hand, the compound stigmasterol (T_3_) showed a minimum inhibition of 0.33 ± 0.06 nm in case No. 55 in *E. coli* (Figure 10B).

The percentages (%) of growth of biofilm formation were calculated after the inhibition of bacterial growth. The results are displayed in Table 4. Among the compounds, 3, 7, 11, 15 tetramethylhexdec-2-en-1-01 (T_2_) has the highest antibiofilm and antimicrobial activities against all bacterial species. The maximum inhibition was 80.10% for case number 17 and 79.74% for case number 55 (Table 4). Investigations across the three compounds of ethyl acetate have shown that 12.5 µL of ethyl acetate demonstrates maximum antimicrobial activity.

## 4. Discussions

Dental plaque samples were collected from 100 individuals with various oral complications patients attending the outdoor department of the dental unit at Rajshahi Medical College and Hospital, Rajshahi, Bangladesh. *Cynodon dactylon* was collected, pulverized, isolated, and purified using column chromatographic techniques; subsequently, biofilms were produced from bacteria grown on MSB medium using a microplate-based system in an in vitro model, which detected the adhesion strength of the biofilm developed by *S. mutans* in the 100 studied samples. It was also revealed that, among these 100 cases, patients with very poor oral hygiene had been suffering from spontaneous bleeding and severe gum diseases and had severe dental plaque, which badly affected most of the teeth of the oral cavity. This study also revealed that the patients had different levels of oral hygiene, such as good, average, and poor. A majority of the patients (61%) had good oral hygiene, while 31% had average and 8% had poor oral hygiene. An almost similar result was found in several experiments where gingivitis was dominant in children, teenagers, and adults, and a maximum of 80% of the global residents experienced mild to moderate gingivitis [33]. There were 242 (48.9%) male and 253 (51.1%) female participants, and most (20.2%) of the participants were six years old [34]. Out of the 100 cases, two (case numbers 17 and 55) were chosen for further analysis because they both had severe plaque scores and poor oral hygiene conditions and showed strong biofilm formation. All the results summarize that patients 17 and 55 had strong biofilm formation. Patient number 17 exhibited the highest level (4.48 ± 0.42 nm) of *S. mutans* biofilm formation. Patient number 55 showed a biofilm formation level of 4.32 ± 0.65 nm. Thus, patients 17 and 55 were selected for biofilm formation.

Among the compounds, 3,7,11,15 tetramethylhexdec-2-en-1-01 has the highest antibiofilm and antimicrobial activities against all bacterial species. The maximum inhibition was 80.10% with a *p*-value < 0.05 in case number 17 and 53.74% in case number 55. Similarly, Sharma and Singh [35] have shown that *C. dactylon* possesses a synergistic effect against *Escherichia coli, Pseudomonas aeruginosa, Staphylococcus coagulasse,* and *Enterococcus faecalis*. *C. dactylon* and other extracts of tested medicinal plants have a broad spectrum of antibacterial activity against a panel of bacteria that cause the most prevalent bacterial illnesses. Investigations across three compounds of ethyl acetate showed that a lower minimum inhibitory concentration was exposed by 3, 7, 11,15 tetramethyl hexadeca-2-en-1-o1 (T2) on *S. mutans* at 12.5 µL/mL. This study agrees with other studies. Adikwu et al. [28] reported that organisms exhibited smaller MIC when exposed to extracts of *Psidium guajava*: an MIC of 3.125 mg/mL was found using methanol extract. Zayed et al. [13] stated that minimum biofilm inhibitory concentrations of alcoholic green tea extracts were in the range of 3.1 to 12.5 mg/mL. Among the compounds,3, 7, 11, 15 tetramethylhexdec-2-en-1-01 (T2) has the highest antibiofilm and antimicrobial activities against all bacterial species. The maximum inhibition was 80.10% in case number 17 and 79.74% inhibition in case number 55. The present experiment is similar to the study of Shekar et al. [36]; they showed 75.54% inhibition of *S. mutans* bacteria. Nallathambi and Bhargavan [37] recorded the existence of bioactive compounds in the moist extract of *C. dactylon* by using a gas chromatograph and a mass spectrometer with a quadruple double-focusing mass analyzer and detected eight compounds which included Alanine, 9, 12-octadecadienoic acid, n-Hexadecenoic acid, oleic acid, 3-octyl-methyl-ester trans, phytoderivatives and coumarine, 3-(2-4-dinitrophenyl), etc. The compounds of phytol derivates are used due to their antimicrobial, anticancer, anti-inflammatory, antioxidant, and diuretic properties [23]. Here, antimicrobial activity was assessed for bacterial growth of inhibition. Sharma and Singh [35] found that *S. sanguis* and *S. mitis* were the most susceptible bacteria to all plant extracts as compared to *S*. *mutans* and *Enterococci*. The reported plant extract showed significant activity against the investigated microbial strain. The present study was similar to that of Sharma and Singh [35]. Similarly, Nalini and Prakasham [38] noticed that *C. dactylon* extract exhibits the simplicity of a zone of hydrolysis, making the process easy, effective, and rapid for the screening of a large number of microorganisms. It was also reported that the extracts of selected species contain a good potential antimicrobial component that can be utilized to prepare potent drugs for treating dental problems and *Streptococcus*-related health.

## 5. Conclusions

The present study was conducted to identify the effect of compounds isolated from *C. dactylon* on the inhibition of *S. mutans* biofilm isolated from dental plaque samples. Three compounds named 3,7,11,15 tetramethylhexadec-2,4 dien-1-o1,3,7,11, 15 tetramethylhexadec-2-en-1-o1, and stigmasterol were extracted. The extracts of *C. dactylon* showed a notable excellent inhibitory impact against particular bacteria strains, including *Escherichia coli, Pseudomonas aeruginosa,* and *Staphylococcus aureus*. The inhibitory activity of these three compounds was evaluated, and the findings demonstrated that 12.5 μL of ethyl acetate extract compound 3,7,11, 15 tetramethyl hexadec-2-en-1-o1 showed the highest antibiofilm activities with an 80.10% growth of inhibition (*p* < 0.05) in case of patient number 17. *C. dactylon* is shown to have surprising health benefits and antibacterial properties. Based on the findings of this study, it could be included that *C. dactylon* possesses several vital antibacterial components that could be very useful in the creation of strong medicines for the treatment of medical conditions. To popularize *C. dactylon* for medicinal purposes, more research is needed.

## Figures and Tables

**Figure 1 biomolecules-13-01292-f001:**
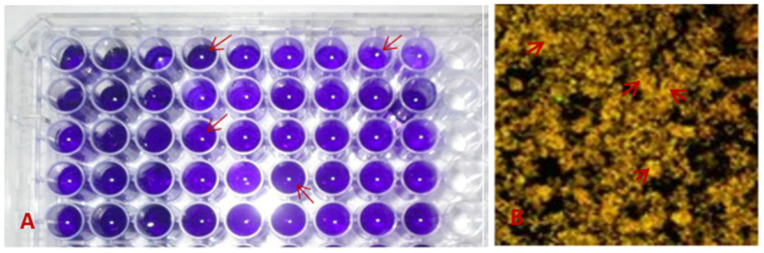
Microtiter ELISA plate showing biofilm formation of the tested bacterial isolates. (**A**) Formation of *S. mutans* bacterial biofilm. (**B**) The cells that adhere to the plate after washing visualized by staining with crystal violet color (by confocal microscopy of *S. mutans*). The arrows show biofilms.

**Figure 2 biomolecules-13-01292-f002:**
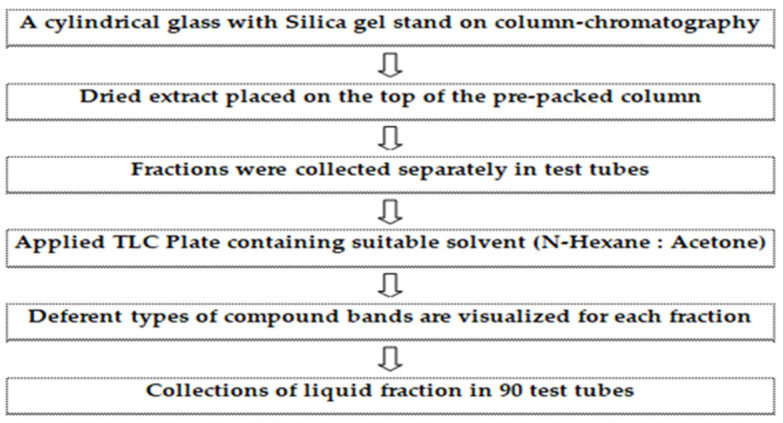
Protocol of isolation and purification of compounds from the medicinal plant (*Cynodon dactylon*) using column chromatographic techniques.

**Figure 3 biomolecules-13-01292-f003:**
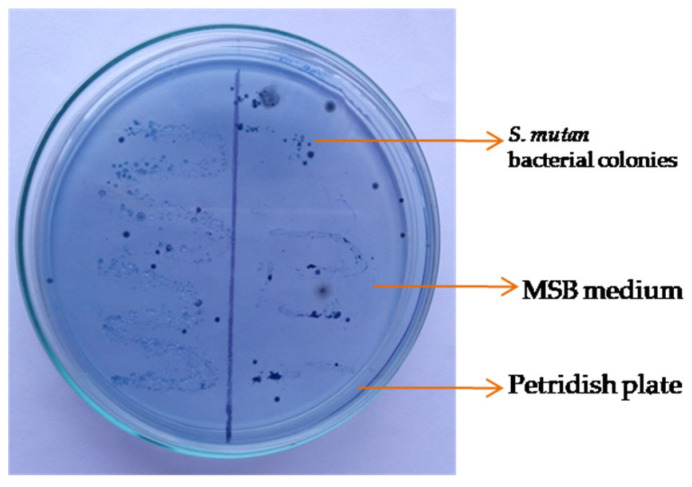
Identification of *Streptococcus* mutans bacteria, grown on MSB selective media overnight at 37 °C.

**Figure 4 biomolecules-13-01292-f004:**
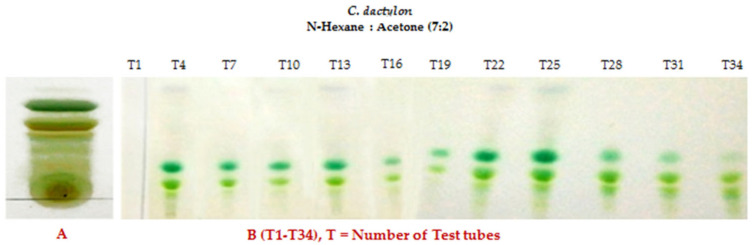
(**A**) Crude extract of *C. dactylon* showing several spots along with three or four prominent brands on the TLC plate after being eluted with 7:2 n-hexane:Acetone- (7 mL and 2 mL) sprayed with vanillin–sulfuric acid reagents during heating. (**B**) TLC plate showing four different bands of *C. dactylon* extracts after finishing the column chromatographic technique and collection of the liquid fraction in test tubes (T1–T34).

**Figure 5 biomolecules-13-01292-f005:**
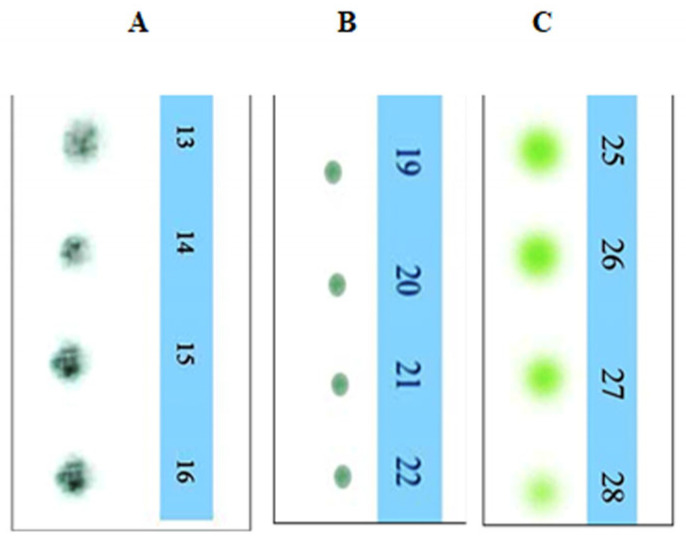
TLC plate showing a single band for each specific compound. (**A**) Single band for 3,7,11,15 tetramethylhexadec-2,4 dien-1-o1, (**B**) single band for 3, 7, 11, 15 tetramethylhexadec-2-en-1-o1, and (**C**) single band for stigmasterol which was confirmed by ^1^H nuclear magnetic resonance (NMR) and C-13 NMR spectrometry at the Department of Microbiology, Jahangirnagar University, Bangladesh.

**Figure 6 biomolecules-13-01292-f006:**
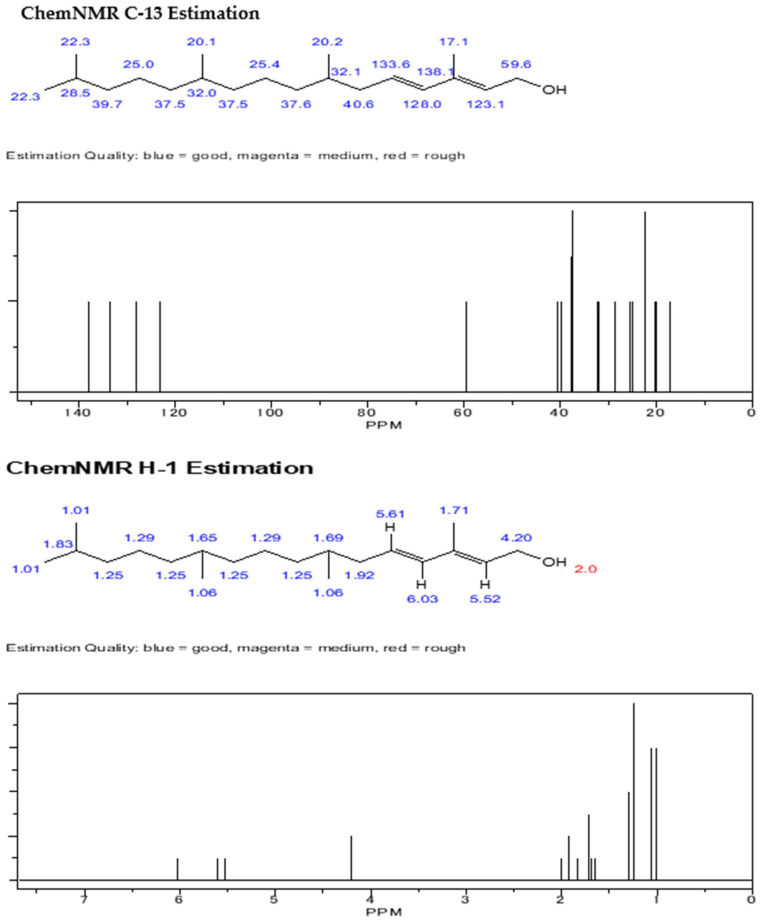
Chemical structure of compound 3,7,11,15 tetramethylhexadec-2,4dien-1-o1 (compound of phytol derivatives). C-13 NMR and 1H NMR spectra of compound 3,7,11,15 tetramethylhexadec-2,4dien-1-o1.

**Figure 7 biomolecules-13-01292-f007:**
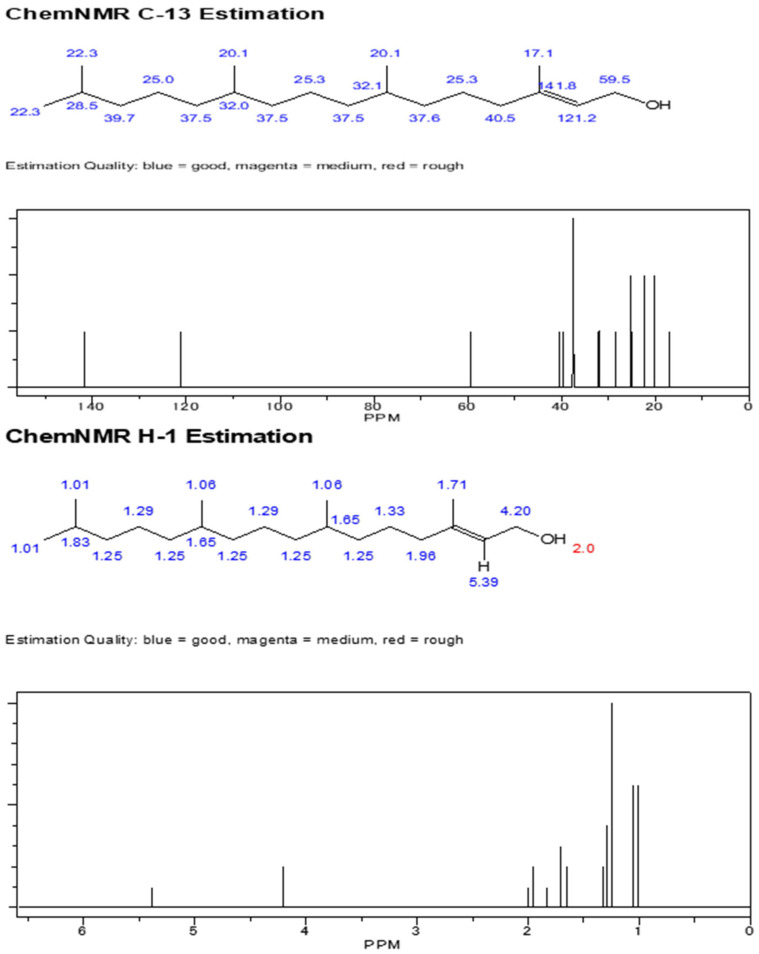
Chemical structure of compound 3,7,11,15 tetramethylhexadeca 2en-1-o1 (compound of phytol derivatives) and C-13 NMR with 1H NMR spectra of compound 3,7,11,15 tetramethyl hexadeca- 2en-1-o1.

**Figure 8 biomolecules-13-01292-f008:**
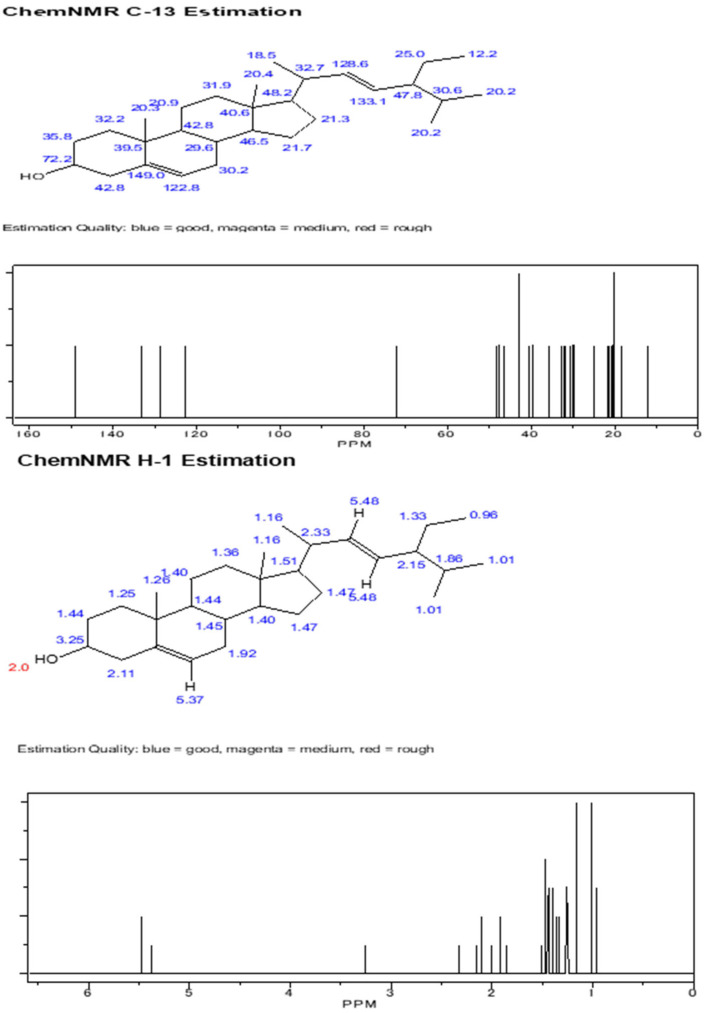
Chemical structure of compound stigmasterol with C-13 NMR and 1H NMR spectra of compound stigmasterol.

**Figure 9 biomolecules-13-01292-f009:**
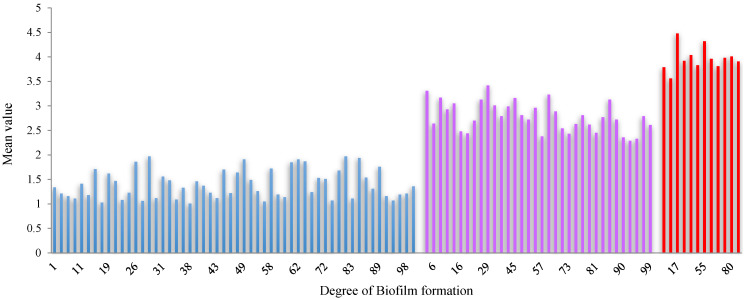
Biofilm formation by the *S. mutans* bacterial plaque sample; data were collected from 100 patients (light blue color indicates Weak, lavender color indicates Moderate, and red color indicates Strong degree of biofilm).

**Figure 10 biomolecules-13-01292-f010:**
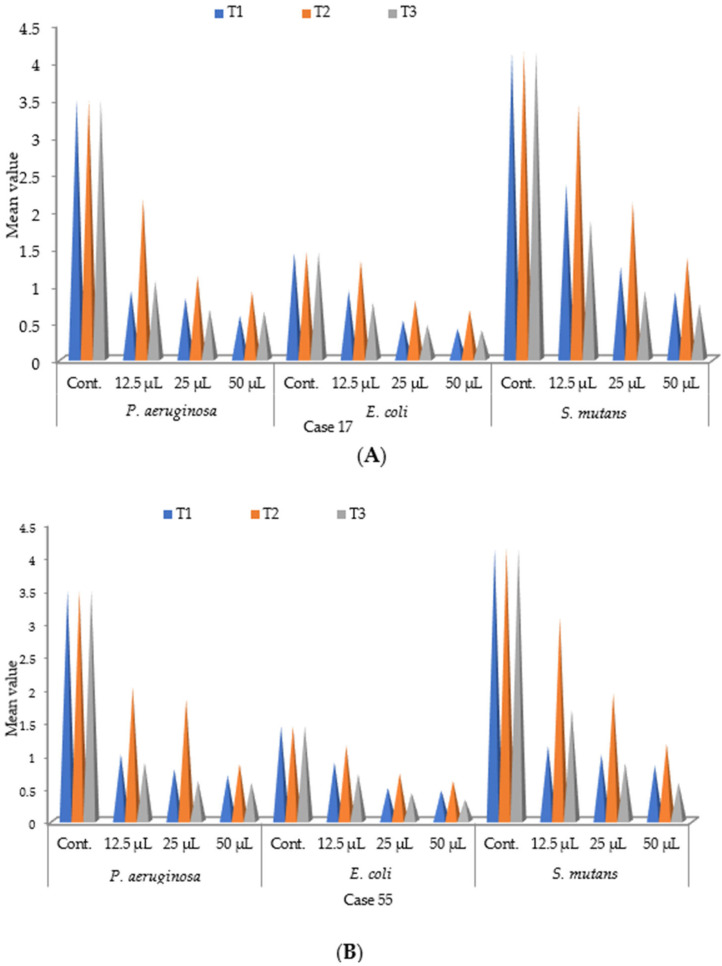
(**A**,**B**). Effect of growth after inhibition of different compounds derived from extract of *C. dactylon* on the biofilm of *P. aeruginosa, E. coli*, and *S. mutans* (Case No. 17 and 55) Control = No treatment; T_1_ = 3, 7, 11, 15 tetramethyl hexadeca-2-4 dien-1-01; T_2_ = 3, 7, 11, 15 tetramethyl hexadeca-2-en-1-01; T_3_ = Stigmasterol.

**Table 1 biomolecules-13-01292-t001:** Compound nature and activity obtained from hot aqua extract of *Cynodon dactylon*.

Name of Compound	Chemical Structure	Molecular Formula	Mol. Weight	R. Time (Min)	Peak Area%
3,7,11,15-tetramethyl-headeca-2, 4 dien-1-o1	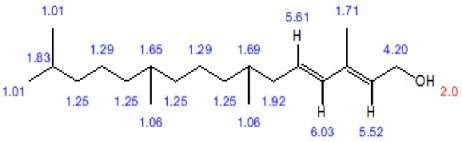	C_20_H_40_	280.54	27.66	80.71
3,7,11,15-tetramethyl-headeca-2-en-1-o1	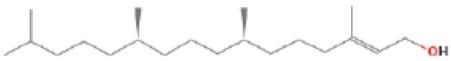	C_20_H_40_O	296.53	25.33	17.10
Stigmasterol	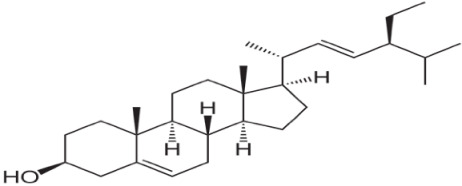	C_29_H_48_O	412.70	20.5	1.64

Source: Dr. Duke’s Phytochemical and Ethnobotanical Databases (https://phytochem.nal.usda.gov, accessed on 10 June 2023). Here, R. time = Retention time; min = minute.

**Table 2 biomolecules-13-01292-t002:** Number of patients and mean values of biofilm formation by the *S. mutans* bacterial plaque.

Degree of Biofilm	Range of OD Values (nm)	Serial Number of Patients	No. of Patients (%)	Mean Value of Biofilm Formation
Weak	0.5–2.0	1,2,5,8,11,12,13,14,19,20,21,22,26,27,28,30,31,34,35,36,38,39,40,42,43,46,47,48,49,51,52,54,58,59,60,61,62,64,66,71,72,75,76,78,83,84,86,87,89,92,95,96,98,100	54 (54%)	1.41± 0.53 c
Moderate	2.0–3.5	3,6,9,10,15,16,18,23,25,29,32,37,41,45,50,53,56,57,65,67,69,73,74,77,79,81,82,85,88,90,91,94,97,99	34 (34%)	2.79± 0.31 b
Strong	3.5–5.0	4,7,17,24,33,44,55,63,68,70,80,93	12 (12%)	3.97± 0.46 a

In a row, different letters represent significant differences at *p* < 0.05, according to DMRT.

**Table 3 biomolecules-13-01292-t003:** Minimum inhibitory concentration (MIC) of *C. dactylon.*

Ethyl Acetate Extract of Compounds	Minimum Inhibitory Concentration (µL/mL)		
	*P. aeruginosa*	*E. coli*	*Streptococcus mutans*
3, 7, 11, 15 tetramethyl hexadeca-2-4 dien-1-01	25.0	50.0	25.0
3, 7, 11, 15 tetramethyl hexadeca-2-en-1-01	25.0	50.0	12.5
Stigmasterol	50.0	50.0	25.0

**Table 4 biomolecules-13-01292-t004:** Percentage (%) of growth after inhibition of different compound extracts of *C. dactylon* on the biofilm of *P. aeruginosa, E. coli,* and *S. mutans* bacteria (Patients No. 17 and 55).

Patient No.	Compound	Bacteria								
		*P. aeruginosa*			*E. coli*			*S. mutans*		
		12.5 µL	25 µL	50 µL	12.5 µL	25 µL	50 µL	12.5 µL	25 µL	50 µL
Patient No. 17	**T_1_**	16.09 a	31.61 b	24.42 ab	14.18 b	35.96 b	29.47 ab	42.96 b	73.42 a	70.50 a
	**T_2_**	20.98 a	70.50 a	34.77 a	27.93 a	47.21 a	38.10 a	54.61 a	80.10 a	76.30 a
	**T_3_**	4.60 b	18.96 c	9.48 b	6.98 c	19.79 c	21.79 c	16.26 b	26.94 b	33.50 b
Patient No. 55	**T_1_**	19.54 b	31.82 b	16.38 b	13.38 b	20.76 b	17.24 b	23.98 b	51.35 a	23.56 b
	**T_2_**	33.62 a	47.41 a	27.23 a	26.17 a	38.52 a	29.05 a	39.81 a	79.74 a	75.69 a
	**T_3_**	12.64 c	19.52 c	12.43 b	9.91 b	16.38 b	11.94 b	16.45 c	30.18 b	19.42 b

In a row, different letters represent significant differences at *p* < 0.05, according to DMRT. T_1_ = 3,7,11,15 tetramethyl hexadeca-2-4 dien-1-01; T_2_ = 3,7,11,15 tetramethyl hexadeca-2-en-1-01 and T_3_ = Stigmasterol.

## Data Availability

The data used in this study are available from the corresponding author upon reasonable request.

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
