# Peer review of "Application of Three Compounds Extracted from *Cynodon dactylon* against *Streptococcus mutans* Biofilm Formation to Prevent Oral Diseases"

_biomolecules, 2023, doi:10.3390/biom13091292_

Round 1
Reviewer 1 Report (Previous Reviewer 1)
In this research, the authors investigated of three compounds extracted from Cynodon dactylon against Streptococcus mutans biofilm formation. The article exhibits a clear and concise writing style, with well-articulated results that support the conclusions drawn. I think this manuscript is well-written. The comments as following as:
1.in this manuscript, two concentrations of 50 µL and 100 µL were selected. please explain it.
2.Please double-check reference style.
Author Response
Please see the attachment

Reviewer 2 Report (Previous Reviewer 2)
1. Figure 1a shows the scan of the 96-well plate, and it is suggested to add the picture of crystal violet staining under light microscope.
Figure 1b does not indicate the type of picture detection (electron microscope or light microscope) in the note, and the picture lacks a scale.
2. Figure 3. Identification of Streptococcus mutans bacteria, grown on MSB selective media 213 for overnight at 37°C .
Figure 3 The pixels are not high enough, and the colonies are small. It is suggested to add local magnification. And the author described it as "Identification of Streptococcus mutans bacteria" in the figure note. It is not reliable to confirm that the colony is Streptococcus mutans bacteria only by morphology, so the author is suggested to change the expression.
3. Considering that the legend of figures should explain what the figure mean instead of containing discussion, please decide whether the blue part of the legend in Figure 5 needs to be deleted or moved.
4. The newly added figures 6-8 are not proportional to the length and width, and they are all related to chemical structure. Please consider whether the three figures can be integrated.
5. Page10 3.3 section formatting error, no first line indent. And the authors are inconsistent in the writing of the cited literature, such as in line294-297:
“Finally, the nomen- 294 clature of these above three compounds was confirmed by comparing it with some data obtained from a previous study done by Upadhya et al. 2014 (31) and Subavathy and Thilaga (2016)(32).”
For the description of the author and year in the citation, it is recommended that it be unified in the full text.
6. The new Figure 10 still does not explain how to get the MIC.
The authors mentioned in the Materials and Methods section that "The lowest concentration of the used compounds that inhibited biofilm formation by 80% was used to calculate the minimum biofilm inhibitory concentration for the tested compounds”
(Page5 line193-194)
However, the process of determining the MIC was not reflected in the experiment. Please explain how the authors determined the MIC of 50μl and 100μl.
It is suggested that the authors supplement the experiments related to MIC concentration determination, and the experimental results are suggested to use the colony growth picture on the plate for more intuitive display.
7. In addition, please explain why there are only two groups of T1 and T2 in Figure 10, but there are two groups of T1, T2 and T3 in Figure 11, and there are three concentrations of 25μl, 50μl and 100μl.
8. The histograms in the article lack bar values, and the figures do not reflect the statistical analysis between groups. It is suggested to optimize and analyze the pictures.
1. Figure 1a shows the scan of the 96-well plate, and it is suggested to add the picture of crystal violet staining under light microscope.
Figure 1b does not indicate the type of picture detection (electron microscope or light microscope) in the note, and the picture lacks a scale.
2. Figure 3. Identification of Streptococcus mutans bacteria, grown on MSB selective media 213 for overnight at 37°C .
Figure 3 The pixels are not high enough, and the colonies are small. It is suggested to add local magnification. And the author described it as "Identification of Streptococcus mutans bacteria" in the figure note. It is not reliable to confirm that the colony is Streptococcus mutans bacteria only by morphology, so the author is suggested to change the expression.
3. Considering that the legend of figures should explain what the figure mean instead of containing discussion, please decide whether the blue part of the legend in Figure 5 needs to be deleted or moved.
4. The newly added figures 6-8 are not proportional to the length and width, and they are all related to chemical structure. Please consider whether the three figures can be integrated.
5. Page10 3.3 section formatting error, no first line indent. And the authors are inconsistent in the writing of the cited literature, such as in line294-297:
“Finally, the nomen- 294 clature of these above three compounds was confirmed by comparing it with some data obtained from a previous study done by Upadhya et al. 2014 (31) and Subavathy and Thilaga (2016)(32).”
For the description of the author and year in the citation, it is recommended that it be unified in the full text.
6. The new Figure 10 still does not explain how to get the MIC.
The authors mentioned in the Materials and Methods section that "The lowest concentration of the used compounds that inhibited biofilm formation by 80% was used to calculate the minimum biofilm inhibitory concentration for the tested compounds”
(Page5 line193-194)
However, the process of determining the MIC was not reflected in the experiment. Please explain how the authors determined the MIC of 50μl and 100μl.
It is suggested that the authors supplement the experiments related to MIC concentration determination, and the experimental results are suggested to use the colony growth picture on the plate for more intuitive display.
7. In addition, please explain why there are only two groups of T1 and T2 in Figure 10, but there are two groups of T1, T2 and T3 in Figure 11, and there are three concentrations of 25μl, 50μl and 100μl.
8. The histograms in the article lack bar values, and the figures do not reflect the statistical analysis between groups. It is suggested to optimize and analyze the pictures.
Author Response
Please see the attachment

Reviewer 3 Report (New Reviewer)
Dear authors.
Diseases of the oral organs are widespread everywhere, caries affects all segments of the population, in all countries of the world. Many plants contain metabolites with biological activities. The study of the mechanisms of action of these metabolites, including inhibitory properties, is an interesting and relevant topic. The scientific content of the manuscript justifies its publication, but some additions and modifications will significantly improve the quality of the article.
Major comments:
Abstract
1) L.28-30, safety studies of the isolated components and clinical trials are needed to confirm this statement.
Materials and Methods
2) L. 125-126, requires explanation.
3) s.2.4, who confirmed the conformity of the collected plants?
4) People took part in the study. The protocol of the ethics committee is not specified.
5) Has the consent of the patients to the collection of biomaterials been obtained?
6) For the software used, authors need to specify full information (developer, year, version, city, country).
7) L.201. The generally accepted confidence probability is 0.05. The authors chose 0.01. This requires an explanation.
8) In the References, 50% of publications refer to 2018-2023 (the last 5 years); the remaining 50% of used sources are older than 5 years. It is recommended to increase the share of references to sources published over the last 5 years when analyzing the current state of research in the area under consideration, since this area of knowledge is rapidly developing.
Author Response
Please see the attachment

Reviewer 4 Report (New Reviewer)
The manuscript investigates the isolation procedure of compounds from Cynodon dactylon for application against Streptococcus mutans biofilm formation to prevent oral diseases. According to the literature review, the research concept and the methods have been applied by several authors, but different kinds of compounds have been subjected to the investigation. This manuscript could represent an added scientific value in the sense of final application. But, first of all, it should be written in a proper form to be suitable for revision. The text is highlighted in blue color, the quality of figures is low as those are included as copy paste and not in a proper resolution range. Moreover, some parts, including the results are misleading. The quantity of the experiments carried out could lead to yield valuable conclusion. In my opinion this manuscript is not suitable for publication in the present form.
Figure 10. Minimum inhibitory concentration (MIC) is given as 50 μL and 100 μL concentration - why not as concentration in mass unit per volume?
Moderate changes are required.
Round 2
Reviewer 4 Report (New Reviewer)
In my opinion, the topic is interesting and the research is appropriate, but the representation of the results is still not satisfactory.
The graphics of some figs are low, fig. 4 which shows TLC plates should be edited or designed in order to fit the appearance of the entire manuscript to the standard of the journal.
The concenrtrations of the MICs have now been given as proposed, but a wider discussion and description of the results should be given.
In my opinion, the topic is interesting and the research is appropriate, but the representation of the results is still not satisfactory.
The graphics of some figs are low, fig. 4 which shows TLC plates should be edited or designed in order to fit the appearance of the entire manuscript to the standard of the journal.
The concenrtrations of the MICs have now been given as proposed, but a wider discussion and description of the results should be given.
Author Response
Please see the attachment

This manuscript is a resubmission of an earlier submission. The following is a list of the peer review reports and author responses from that submission.
Round 1
Reviewer 1 Report
(1) The introduction part is a bit confusing and should be arranged logically.
(2) Tile is “Application of three compounds extracted from Cynodon 2 dactylon against Streptococcus mutans biofilm formation to 3 prevent oral diseases”. It implies that this study focuses on the effect of Cynodon 2 dactylon on the biofilm formation. However, the material and results part, there is only one experiment about biofilm formation. The understanding of the overall biofilm formation process depends on the deep understanding of the main aspects regulating biofilm development, such as the initial adhesion, biofilm development. However, in this study, effect of three compounds on initial adhesion and biofilm development was not detected.
(3) If the authors said these compounds could be applied to prevent oral diseases, more animal models designed for biofilm formation in vivo study should be used.
(4) It may be more readable to apply figures to illustrate the biofilm formation difference among the 100 samples .
(5) Please illustrate why 3, 7, 11, 15 tetramethyl hexadec-2-4dien 1-o1, compound 3, 7, 11, 15 tetra- 22 methylhexadec-2-en-1-o1 from phytol derivatives, and stigmasterol from Cynodon 2 dactylon were chosen. Does Cynodon 2 dactylon contain these componds in high level? It found Tricosane (22.05 %), 1, 2-Propanediol (20.30%), 3-benzyloxy-1, 2-diacetyl (12.62%) were present at maximum level in Cynodon dactylon l in some paper(Kaleeswaran B., Ilavenil S., Ravikumar S. Screening of phytochemical properties and antibacterial activity of Cynodon dactylon L. International Journal of Current Research. 2010;3:83–88). How about these three compounds used in this study?
(6) Dental plaque samples were collected from 100 individuals. However, only 2 samples were used to detect the biofilm formation. It may be more convincing to classify 100 samples in different degrees such as weak/ moderate/ strong producers according to the amount of plaque formation and then select some samples for each degree for subsequent experiments.
(7) I suggest to add Statistical Analysis part.
(8) Please double check the reference style required by the journal.
Author Response
Dear Reviewer,
Thank you for your critical review of our manuscript. Attached, please find our responses to your review.

Reviewer 2 Report
This study aimed to determine the antimicrobial and antibiofilm effectivity of different compounds of C. dactylon against the biofilm of some pathogenic bacteria. The objective of the paper is clear, but there are still some problems in the experimental design that need to be improved, and the figures and tables can also be improved. Please see below for details.
1. Page 5: Figure2A The picture is distorted. It is recommended to adjust the ratio of length to width. The author's comment of Figure2B is "Cylindrical glass 161 column containing silica gel with liquid solvent and sample extraction", but the current picture is not focused. Fuzzy, recommend replacement. Figure2C is just a simple digital display, which is of little significance. It is suggested to replace it.
2. Page 6: Figure 3: Identification of Streptococcus mutans bacteria on MSB selective media, Considering the author's description of the colony as "hard, raised, convex pale blue colonies with a frosty glass appearance", it is suggested to add a high-magnification picture here to show the colony morphology.
3. Page 6: 3.2 contains too little content, so it is suggested to appropriately add the description of the resulting data. Figure 3 also has the problem of distortion, and it is suggested to adjust the aspect ratio.
4. Page 7: The unit of adhesion strength is not indicated, and the measurement method of adhesion strength is not indicated in the previous paper. It is suggested to use the picture instead of the content of the table, which is more intuitive.
5. Page 8: The authors applied these three compounds to bacterial biofilms at different concentrations. The conclusion is "compounds 3, 7, 11, 15 tetramethylhexdec-2-en-1-01 has the highest 236 antibiofilm and antimicrobial activities against all bacterial species ". Please explain how to determine the applied drug concentration of the three compounds and whether the antimicrobial properties of the drugs are comparable under different concentrations.
6. It is suggested that the author supplement the determination experiments of the minimum inhibitory concentration MIC and the minimum bactericidal concentration MBC of the three kinds of bacteria by three compounds respectively.
7. Page 9: The contents of the table are cluttered at present. It is suggested that the author consider replacing the table with pictures.
8. Page 10: The author mentioned that "C. dactylon is shown to have amazing health benefits and antibacterial properties." This statement is too absolute. This study is only in vitro experiment, and the setting of drug concentration needs to be studied. It is suggested to change the expression here and add in vivo experiments for verification.
9. In this experiment, the author only explored the antibacterial performance of C. dactylon, and did not consider the biosafety of C. dactylon under the treatment dose. It is suggested to add related cell biology experiments to verify the biosafety of the drug.
10. It is suggested to add the DISCUSSION part to discuss and analyze the experimental content.
11. Among the 32 references, 15 are from five years ago. It is suggested to update the references
12. It is suggested to add an appendix document to clarify the dental disease information of each patient included in the experiment. Considering that the author needs to collect materials from the clinic, please add the ethical number of the experiment. the author mentioned that "Among all the 100 cases, two (case numbers 17 and 55) were selected for further analysis." Please explain the reasons for choosing 17 and 55.
Author Response

(The authors gave the same response as above.)

Round 2
Reviewer 1 Report
Dear authors,
Thank you for addressing most issues raised in the review.
Perhaps it would be more readable to move Table 1 to the Appendix and provide a summary of the raw data in the main manuscript.
Author Response
Dear Reviewer,
Thank you for your review of our manuscript. Please see the attachment.
Best regards
Aminur Rahman

Reviewer 2 Report
We found that the author has made some modifications according to our opinions, but there are still some experiments that are not supplemented.
We mentioned some questions about the new manuscript and suggested that the authors could conduct some supplementary experiments.
1. We think that figure 1 might be of little significance. Since we cannot directly see the bacterial biofilm through the scan picture of 96-well plate, it is suggested to add the electron microscope image.
2. It is suggested to modify Figure 2. The text content in the flow chart should be concise and clear.
3. The current figure 3 has low pixel, unclear and chaotic background. It is suggested to change the picture.
4. Please confirm whether "2.7. Statistical analysis" needs to be changed to italics?
5. Figure4 also has the problem of low pixel and unclear. The author thinks that figure4 is "a high-magnification picture", but we believe that it is a scanning picture of the medium. Besides, the monoclonal strains isolated from the right side of the medium in Figure 4 are dark blue. Please check if the description of "pale" is accurate.
6. Figure 6 also has the problem of length and width mismatch, which is suggested to be adjusted.
7. In your cover letter, The authors mention that "We have used different concentrations (25 µl, 50 µl, 100 µl) of extracts to protect against biofilm formation. The 50 µl concentration showed better results. "However, the author did not explain how to determine the experimental concentration of three drugs and set the concentration interval. Please give a clear explanation.
Besides, we believe that adding three compounds is necessary for the minimum inhibitory concentration MIC and minimum bactericidal concentration MBC experiments of different bacteria. It is suggested that the author add relevant experiments, and the results of the experiments are suggested to display using pictures of colony growth on the plate and bar charts instead of tables.
8. The contents of table1 are suggested to be placed In the appendix file instead of the main body. In addition, the author mentioned that "In terms of dental plaque, mild, moderate and severe scores were obtained".
"patients were under different oral hygiene agreement such as good, average and poor"
However, it does not give the definition of specific classification, and it is suggested to add a explanation.
9. Considering that this experiment involves the collection of clinical samples, it is suggested to add the corresponding ethical number, which we think is necessary.
10. The author mentioned in the conclusion that "C. dactylon is shown to have amazing health benefits and antibacterial properties." This sentence is too absolute and not appropriate enough, we recommend authors to modify the sentences.
11. In addition, relevant experiments on the biological safety of drugs have not been added in this study. Considering the author repeatedly mentions the purpose and prospect of the medical application of the compound, we believe that biosafety-related experiments are necessary. We advise authors to do supplementary experiments.
12. We find that the contents of the discussion section are similar to the contents of previous sections. For example,
“Children and teenagers were 835 selected in a more significant number and adult patients in a smaller number for this study 836 as the WHO (World Health Organization) reported that 60 to 90% of children and young 837 adults affected by dental caries have S. mutans bacterial biofilm to a greater extent than 838 adults [29].”(page 3)
“Children and teenagers were selected in a larger number and 149 adult patients in a smaller number for this study as the WHO (World Health Organiza- 150 tion) reported that 60 to 90% of children and young adults affected by dental caries have 151 S. mutans bacterial biofilm to a greater extent than adults [29].” (page 11)
We believe that the discussion section should not be a simple result repetition. We suggest authors to rethink and sort out logic in order to improve the depth of the discussion section.
Author Response
Dear Reviewer,
Thank you for your review of our manuscript. Your critical review has improved our manuscript. Please see the attachment.
Best regards
Aminur Rahman
